# A Multitemporal Mountain Rice Identification and Extraction Method Based on the Optimal Feature Combination and Machine Learning

**Kaili Zhang** [1], **Yonggang Chen** [1,*], **Bokun Zhang** [2], **Junjie Hu** [2] and **Wentao Wang** [1]

1   College of Environmental and Resource Sciences, Zhejiang A&F University, Hangzhou 311300, China
2   Zhejiang Engineering Geophysical Prospecting and Design Institute Co., Ltd., Hangzhou 310005, China
*   Correspondence: cyggis@zafu.edu.cn

**Abstract:** The quick and precise assessment of rice distribution by remote sensing technology is important for agricultural development. However, mountain rice is limited by the complex terrain, and its distribution is fragmented. Therefore, it is necessary to fully use the abundant spatial, temporal, and spectral information of remote sensing imagery. This study extracted 22 classification features from Sentinel-2 imagery (spectral features, texture features, terrain features, and a custom spectral-spatial feature). A feature selection method based on the optimal extraction period of features (OPFSM) was constructed, and a multitemporal feature combination (MC) was generated based on the separability of different vegetation types in different periods. Finally, the extraction accuracy of MC for mountain rice was explored using Random Forest (RF), CatBoost, and ExtraTrees (ET) machine learning algorithms. The results show that MC improved the overall accuracy (OA) by 3–6% when compared to the feature combinations in each rice growth stage, and by 7–14% when compared to the original images. MC based on the ET classifier (MC-ET) performed the best for rice extraction, with the OA of 86%, Kappa coefficient of 0.81, and F1 score of 0.95 for rice. The study demonstrated that OPFSM could be used as a reference for selecting multitemporal features, and the MC-ET classification scheme has high application potential for mountain rice extraction.

**Keywords:** mountain rice; separability; feature selection; machine learning; multitemporal feature combination; Sentinel-2

## 1. Introduction

In recent years, accurate monitoring of crop acreage and distribution has been crucial for economic development and food security under both climate change and epidemic influences. However, traditional methods for determining crop distribution, such as expert knowledge or field measurements, are limited by complex environmental and human interference, with high costs and low efficiency [1]. Meanwhile, remote sensing technology can provide information on spatial continuity and has the advantage of rapid, large-scale observations. It, therefore, provides a cost-effective method to monitor crop distribution [2,3].

Current research on crop distribution monitoring using remote sensing technology can be divided into two main categories based on the sensor's working mode: using synthetic aperture radar (SAR) to obtain the electromagnetic radiation and scattering properties of targets [4,5], and using optical satellites to acquire the spectral information on targets [6,7]. Evans et al. [8] used a dual-season set of fine-spatial-resolution C-band (RADARSAT-1) and L-band (JERS-1) SAR imagery to effectively avoid the problem of cloud cover and the influence of dense vegetation canopy. In such a way, they determined the distribution of the variety of habitats in the Lower Nhecolândia subregion of the Pantanal at a regional scale. Martone et al. [9] concluded that SAR sensors are weather and daylight independent for reliable mapping and monitoring of forest areas. By applying them, they created the first

global forest/non-forest classification map from the TanDEM-X single-pass interferometric SAR (InSAR) dataset. Using radar remote sensing can effectively reduce the impact of clouds on monitoring accuracy, especially for mountain crops, and data sources such as ENVISAT ASAR, Sentinel-1, and POLARSAR are suitable [10,11]. However, it should be noted that radar has background noise and system noise when receiving signals. High noise levels can weaken the signal if not suppressed effectively, thus affecting detection performance [12]. Furthermore, due to the side-looking nature of radar and the complex mountainous terrain, radar imagery is prone to geometric and radiometric aberrations, and slopes back toward the radar may form shadows [13]. In addition, the phenomena of Foreshortening and Layover [14] can also disturb monitoring. When monitoring crops with optical satellites, images of fixed periods are prone to spectral confusion. Plus, they are also prone to inadequate feature description under complex weather conditions in mountainous areas. However, where spectral information is insufficient, phenological information can be an important supplement [15,16]. Crops exhibit different characteristics at different growth stages, and using multitemporal images to extract crops is an effective and widely used idea in research. For example, d'Andrimont et al. [17] proposed a normalised yellow index to determine the spatial distribution of rape based on the colour variation of rape flowers (colour characteristics are strongest during flowering and weakest during withering). Biradar et al. [18] quantified the area and spatial distribution of farmland in India using multitemporal MODIS images based on the normalised difference vegetation index (NDVI), enhanced vegetation index (EVI), and surface water index (LSWI), demonstrating the potential of phenology algorithm in multi-season planting. Sibanda et al. [19] successfully distinguished cotton from maize and sorghum using 16-day time series of moderate-resolution imaging spectroradiometer-normalised difference vegetation index (MODIS NDVI) data. In general, low-resolution imagery is often used in areas with large planting scales and fewer clouds. Mountain crops are limited by terrain and climatic conditions, and the fields present small-scale, fragmented and scattered characteristics. Low-resolution imagery is not suitable for the classification and extraction of fine crops. Therefore, a suitable image source must be found before the study begins, which can create a problem. In that context, the multispectral imager (MSI) of Sentinel-2 incorporates three additional spectral bands in the red-edge (RE) region, which is expected to improve the monitoring of vegetation information. Sentinel-2 imagery from the European Space Agency (ESA) provides multispectral information with short replay periods, high spatial resolution, and global coverage. The data have been widely used for agricultural and forestry monitoring, as well as crop yield estimation [20,21]. They also open up research opportunities for crop extraction using temporal information.

Use of spectral and phenological information to extract crops still faces a hurdle: the diversity of features cannot be guaranteed for complex land types. Haralick et al. [22] proposed that texture features can describe the structural properties of the object surface well. Zhou et al. [23] suggested that terrain features can highlight terrain differences. They demonstrated that terrain features could also improve the classification accuracy in their experiments on greenhouse shed extraction. In addition, it should be noted that mountain fields are fragmented, and the boundaries of different land types are diverse. Many researchers have demonstrated that using spectral, texture, and terrain features alone does not make the classification results reliable for mixed pixels at the boundary [24–26]. Furthermore, some researchers have found that using the spectral-spatial feature can significantly improve the separability between land types [27–29]. However, the other challenges are how to integrate temporal and feature information adequately, as well as how to avoid feature redundancy and ensure the importance of features. Zhang et al. [30] entered the features of all periods into Random Forest (RF) with out-of-bag (OOB) data error to calculate the importance of features, and selected the top 28 features with importance greater than 0.8 to participate in tea plantation extraction. Although this method has achieved high accuracy in classification, screening all features from each period will lead to a serious 'feature skew' phenomenon, as the same feature may have high importance

in different periods. As a result, the heterogeneity of features is reduced. Therefore, it is also necessary to consider the heterogeneity of features when selecting features from different periods.

When extracting mountain crops from the perspective of feature selection, selecting an image with an open, short monitoring cycle and high resolution is a vital prerequisite for extracting large-scale crops [31]. Ensuring the diversity and heterogeneity of features is an essential guarantee for improving accuracy [32]. Establishing a stable feature combination and classification scheme is an important objective. However, most studies have used spectral information when extracting crops using multitemporal images. In addition, most feature selection methods have focused on the level of feature importance. Adequate integration of temporal dominance and multi-type feature information is necessary for accurate extraction of mountain crops [33]. Rice is the main grain crop in China, and the south of China is the main region for rice cultivation, with mostly hilly and mountainous terrain. This study mainly extracted mountain rice. In order to select a suitable image, this study weighed the advantages and disadvantages of various data in a mountainous environment and finally determined that Sentinel-2 imagery was appropriate. Moreover, spectral features, texture features, terrain features, and a custom spectral-spatial feature were chosen to ensure the diversity of features. More critically, this study constructed a feature-selection method based on the optimal extraction period of features (OPFSM). Using the OPFSM can solve the problem of feature selection in different periods. It can also ensure diversity, heterogeneity, and importance of features while maintaining the separability of vegetation. In addition, different machine learning classifiers and comparison experiments were used to verify the stability of the feature-selection results. By extracting mountain rice using a multitemporal, multi-feature, and multi-classifier approach, this study aimed to explore the optimal feature combination and classification scheme applicable to mountain rice.

## 2. Materials and Methods

### 2.1. Study Area

The study area was part of Sandu Town, Jiande City, Zhejiang Province, China, covering 9.72 km$^2$ between 119°35′E to 119°38′E and 29°31′N to 29°34′N (Figure 1). Sandu Town is the third of the strong agricultural towns at the provincial level in Zhejiang Province. The region is primarily mountainous and hilly, and forest cover is about 75%. The climate is warm and humid, with an average annual temperature of 16.7 °C. Rainfall is plentiful, with an average annual precipitation of 1600 mm and 164 rainy days. Clouds are mostly concentrated in the plum rain season and are relatively abundant in June and July, influenced by rainfall and the monsoon. Rice, tea plantations, and citrus are the main crops. Rice is the main grain crop. Due to their excellent natural conditions for the soil, climate, and water quality, the first batch of Yuan Longping ecological rice planting sites in Zhejiang Province was set up in the study area and successfully trialled in 2021. The mu yield is about 200 to 300 kg higher than ordinary rice. Moreover, there are plans to continue expanding the "giant rice" area planted throughout the town in 2022. The success of the trial planting of ecological rice is significant in helping farmers to increase their yields and incomes and contribute to shared prosperity. It also offers great market prospects for ecological rice cultivation in mountainous regions.

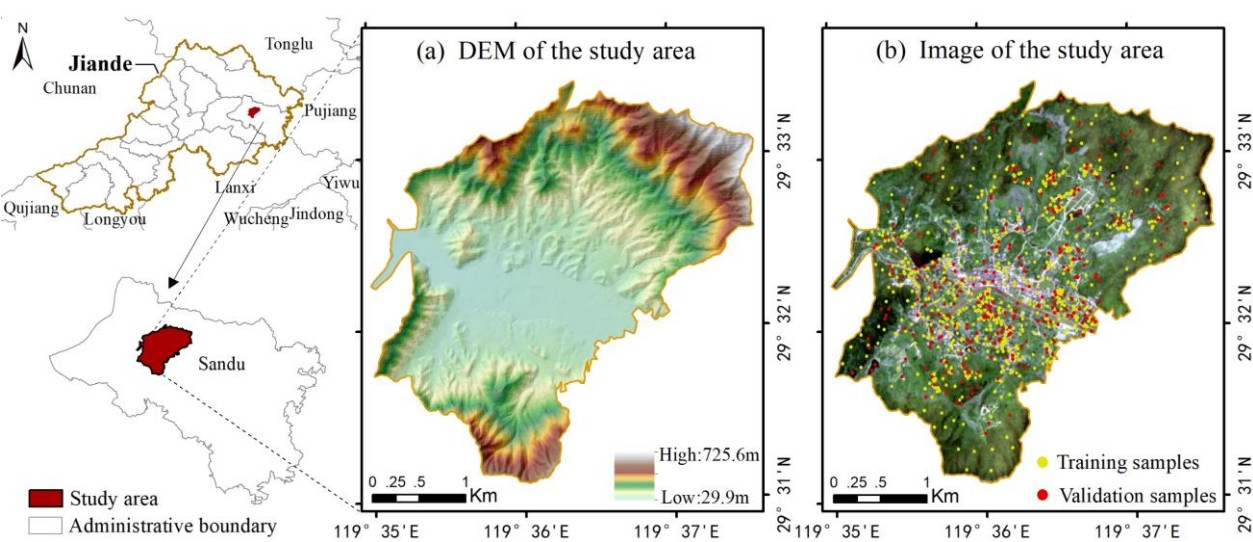

**Figure 1.** Location of the study area and sampling points.

*2.2. Data*

2.2.1. Image Data

Sentinel-2 remote sensing images were obtained from the official public website (https://scihun.copernicus.eu/ (accessed on 9 June 2022)) of the European Space Agency (ESA). The data set covers 13 spectral bands (b1~b12), from visible to shortwave infrared, and with spatial resolutions from 10 m to 60 m. In addition, the Sentinel-2A imagery is the only data with three bands in the red-edge range. It can provide sensitive spectral information in vegetation monitoring studies, effectively improving vegetation classification accuracy, and has great potential for applications in vegetation extraction and biomass prediction.

According to a survey, the majority of rice grown in the study area is single-season. In addition, according to the growth characteristics of local rice (Table 1), the seeding period is from mid-April to mid-May, the growth period is from early June to mid/late July, and the maturity period is from early September to mid-October. At different growth stages, there are obvious spectral reflectance changes in different bands of rice [34]. Therefore, we analysed the cloud coverage of images in recent years based on the growth cycle and spectral reflectance properties of rice and finally selected four Sentinel-2A data (3 May 2021, 28 June 2021, 22 July 2021, 25 September 2021) as the images of rice during the seeding, joining, heading, and maturity periods, respectively. Among them, the image quality is relatively high (less than 10% cloud cover) in the seeding, heading, and maturity periods. However, the images contain partially aggregated clouds in the joining period. The clouds are mainly distributed above the forest, and the area is small. Therefore, this study neglected the change in land type. Only the radiation difference was considered to reconstruct the image by using the image of the heading period to complement this area. Moreover, the selected images were atmospherically corrected based on the L1C level. The L1C level data were orthorectified and geometrically corrected. In addition, the Sentinel-2A images were resampled to 10 m using the SNAP software provided by the ESA.

**Table 1.** Rice growth stages and selected images.

| Growth Stage | Agricultural Stage | Agricultural Time | Imaging Date |
|---|---|---|---|
| | Nursery | Mid-April | |
| Seeding | Emergence | End of April | 3 May 2021 |
| | Transplanting | Early to mid-May | |

**Table 1.** *Cont.*

| Growth Stage | Agricultural Stage | Agricultural Time | Imaging Date |
|---|---|---|---|
| Growth | Tiller | Early June | 28 June 2021 22 July 2021 |
| | Jointing | End of June | |
| | Booting | Early July | |
| | Heading | Mid to late July | |
| Maturity | Maturity | Early September to mid-October | 25 September 2021 |

### 2.2.2. Ground Survey Data and Sample Datasets

The crops are rice, tea plantations, and citrus in the study area. The demand for fruit in all seasons has led to some citrus being cultivated in greenhouses, and agricultural techniques have extended the maturity period of citrus. This makes the temporal features of citrus diverse and unstable. As a result, this paper selected three vegetation (rice, tea plantations, and forest) and three non-vegetation (water, residential, and transportation) factors for study from the perspective of the long-term stability of the crop growth cycle. This study collected samples primarily by field surveying, supplemented by GoogleEarth sub-metre images. In the field survey, rice and tea plantation samples were mainly collected. All samples were kept at least 10 m away from the surrounding boundary to preserve their cleanliness. Among them, there were 358 rice samples and 165 tea plantation samples. In addition, forest, water, residential, and transportation samples were extracted from GoogleEarth sub-metre images. A total of 904 samples were collected, and they were randomly divided into training and validation samples in a ratio of 7:3 (Figure 1).

### 2.2.3. Digital Terrain Data

The study selected 12.5 m resolution DEM data (https://search.asf.alaska.edu/#/ (accessed on 1 July 2022)) to extract terrain features. The data were collected by the Phased Array L-band Synthetic Aperture Radar (PALSAR) of the Advanced Land Observing Satellite (ALOS, launched in 2006).

### 2.3. Methods

This study first constructed initial features (spectral features, texture features, terrain features, and a custom spectral-spatial feature) based on DEM and multi-period images, preferring the initial features of each period, which could form the feature combinations of corresponding periods, i.e., single-temporal feature combinations (SCs). Secondly, the spectral differences between vegetation and non-vegetation were analysed, and the optimal extraction period with the highest separability of vegetation was identified in each feature based on samples of land types. Only the features of the optimal extraction period were involved in feature selection. This process generated a multitemporal feature combination (MC). Finally, the rice extraction accuracies of MC and SCs were compared using different machine learning algorithms. The technical flow of the study is shown in Figure 2.

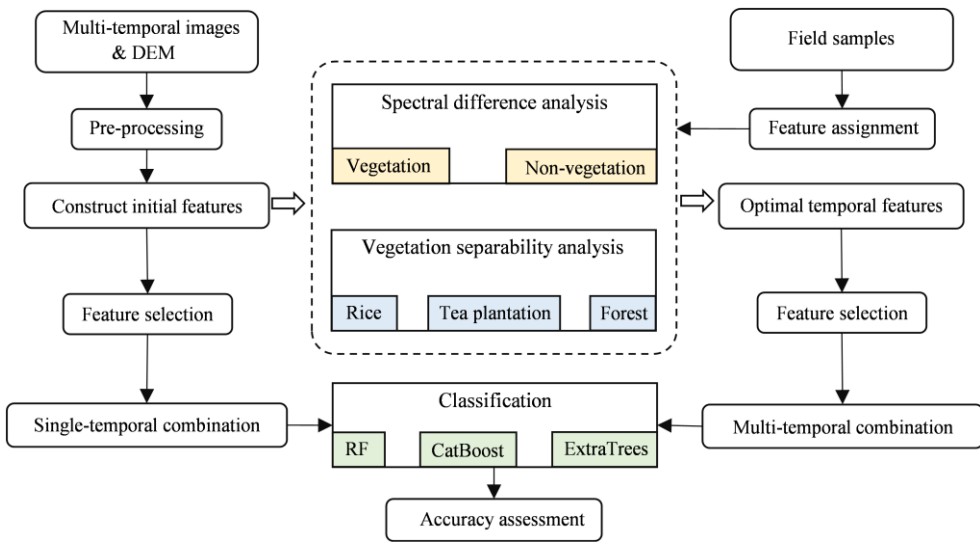

**Figure 2.** Flow chart of the research methodology.

2.3.1. Feature Extraction

This study extracted 13 spectral features, four texture features, four terrain features, and a custom spectral-spatial feature from Sentinel-2A images of different periods (Table 2) [35–39]. Among them, the spectral features were mainly vegetation indices that effectively distinguish rice from other vegetation. In addition, the remote sensing images were transformed by principal components separately. The first principal component (PCA1) mainly reflects the overall albedo of the pixel, and the second principal component (PCA2) mainly reflects the slope change of the spectral curve. The two principal components contain more than 96% of the information and can be used for data compression and classification. Therefore, this study used PCA1 and PCA2 as two spectral features. The texture features were selected from those commonly used in the grey-level co-occurrence matrix [22], extracted in PCA1. Terrain features only reflect differences in terrain distribution and do not account for temporal variations.

**Table 2.** List of features.

| Feature Category | Specific Features |
| --- | --- |
| Spectral features | Chlorophyll Absorption Ratio Index (CARI), Ratio Vegetation Index (RVI), Difference Vegetation Index (DVI), Enhanced Vegetation Index (EVI), Perpendicular Vegetation Index (PVI), Normalised Difference Vegetation Index (NDVI), Soil Adjusted Vegetation Index (SAVI), Normalised Difference Vegetation Index Red-Edge3 (NDVI$_{re3}$), Normalised Difference Water Index (NDWI), Land Surface Water Index (LSWI), Normalised Difference Built-up Index (NDBI), First Principal Component (PCA1), Second Principal Component (PCA2) |
| Textural features | Mean, Homogeneity (HOM), Entropy (ENT), Correlation (COR) |
| Terrain features | DEM, Slope, Aspect, Curvature |
| Spectral-spatial features | Pixel Neighbourhood Similarity Index (PNS) |

In addition, this study constructed a spectral-spatial feature, namely the pixel neighbourhood similarity (PNS) index, from the perspective of a multivariate spectral similarity measure [40]. This was used to determine the similarity between adjacent pixels. As shown in Figure 3, the PNS index was calculated in a 3 × 3 neighbourhood window. The cosine similarity [41] between the centre pixel and the neighbourhood pixels of the window was calculated, and the average result was returned to the central pixel. In this step, the multidimensional spectral information was compressed into a one-dimensional space, and the continuity of the spectral information was ensured. This not only reduced the spectral dimensionality but also ensured the continuity of the spectral information. At the same

time, the spatial distance relationship was combined in the spatial dimension to assign importance weights to neighbourhood pixels. Based on the above process, a spectral-spatial feature with spectral continuity and a spatial relationship was obtained.

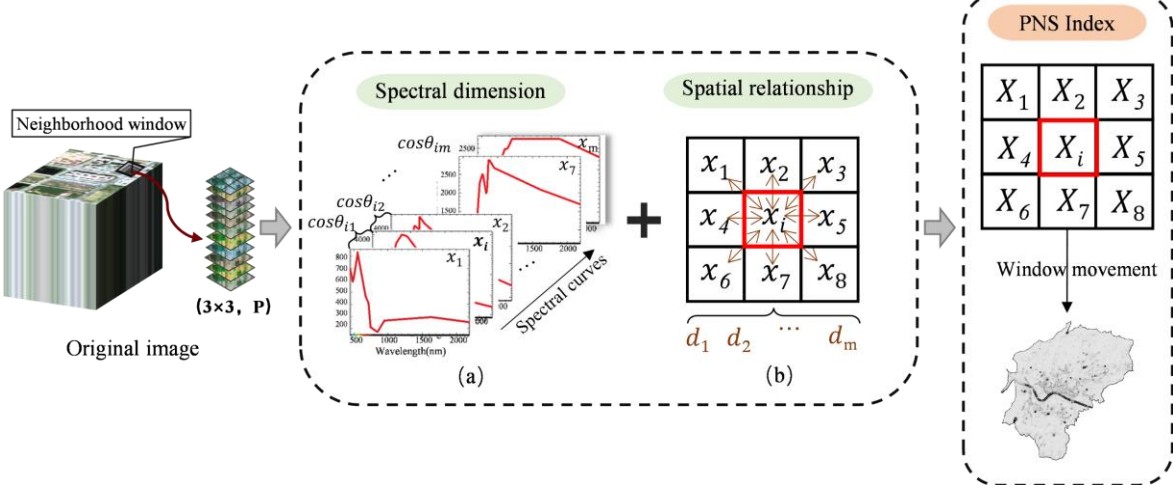

**Figure 3.** Construction process of the PNS Index.

The index is calculated as:

$$PNS = \frac{1}{m} \sum_{n=1}^{m} \frac{R}{d_n} \frac{\sum_{k=1}^{P} x_{ik} \cdot x_{nk}}{\sqrt{\sum_{k=1}^{P} x_{ik}^2 \cdot \sum_{k=1}^{P} x_{nk}^2}} \tag{1}$$

where $i$ is the centre pixel with $m$ neighbouring pixels, $d_n$ is the distance of each neighbouring pixel to $i$, and $\frac{R}{d_n}$ is the relative distance weight of each neighbouring pixel to $i$. Here, $R$ is the spatial resolution of the image. The closer the PNS index is to 1, the higher the similarity between the pixel and the neighbouring pixels, and the more likely it is to be the same type.

2.3.2. Feature Selection

In order to ensure diversity and heterogeneity of feature-selection results, this study constructed a feature-selection method based on the optimal extraction period of each feature (OPFSM). The procedure was broken down into two key steps: determining the optimal extraction period of each feature (A), and using those features in the selection algorithm (B).

(A)   Determine the optimal extraction period for each feature

One-way analysis of variance (ANOVA) [42] is a statistical method used to analyse the difference between the test results (X) and single or multiple influencing factors (Y). It can use the effect size (ES) to measure the magnitude of the difference. This study used one-way ANOVA to judge the separation of different vegetation types by features, and used ES to measure the magnitude of separation. The larger the ES, the higher the separability between vegetation, so that the optimal extraction period of each feature can be determined.

The specific steps of the one-way ANOVA analysis are as follows: (1) Group Y according to X and use normality and homogeneity of variance tests to check whether the data have normality in the overall distribution and whether the $p$-value is significant (less than 0.05 or 0.01). (2) If the data have a normal distribution, but the $p$-value is not significant, the original hypothesis (original hypothesis: variance homogeneity is satisfied) is rejected. This indicates that the data fluctuate inconsistently. In this case, Y can be analysed based on the mean $\pm$ standard deviation to check whether the $p$-value is significant. (3) If the $p$-value is significant at this time, there is a significant difference between the different vegetation

types in a particular feature, and the magnitude of the difference can be further analysed according to the ES indicators.

Two ES indicators, partial $\eta^2$ and Cohen's f, were used in this study to measure the magnitude of separability between vegetation. The partial $\eta^2$ indicator is an independent variable that can explain the size of the overall variance variation of the dependent variable. When using partial $\eta^2$ to represent the effect size, the thresholds for small, medium, and large are 0.01, 0.06, and 0.14, respectively. When using Cohen's f to represent the effect size, the thresholds for small, medium, and large are 0.1, 0.25, and 0.40, respectively. The partial $\eta^2$ and Cohen's f are defined as:

$$\text{Partial } \eta^2 = \frac{SS_A}{SS_A + SS_E} \tag{2}$$

$$\text{Cohen's f} = \sqrt{\frac{\text{Partial } \eta^2}{1 - \text{Partial } \eta^2}} \tag{3}$$

where $SS_A$ represents the between-group variation caused by factor A, and $SS_E$ is the within-group variation.

(B)    Feature-selection algorithm

The correlation-based feature-selection (CFS) algorithm is the classical filtered method based on search strategies [43]. The core of CFS uses a heuristic search strategy and a correlation assessment function to evaluate the value of a feature subset. The heuristic strategy assumes that a good feature subset contains highly correlated features with the classification targets, but features are not correlated with each other. In this paper, the best first search strategy was used to search for features in the CFS algorithm, with the search direction set to forward selection and the initial feature subset S set to empty. The selection process is shown in Figure 4. The feature estimates (represented by the merit value) were calculated according to the feature–class and feature–feature relationships. The feature with the largest merit value was selected to S, and the next largest feature was chosen. Suppose the merit value of these two features was less than the previous merit. In that case, the second feature was removed, and so on, until the largest merit combination was identified as the optimal feature combination. The merit value is calculated by:

$$\text{Merit}_s = \frac{k\overline{r_{cf}}}{\sqrt{k + k(k-1)\overline{r_{ff}}}} \tag{4}$$

where $k$ is the number of features contained in the combination S; $\overline{r_{cf}}$ indicates the average correlation between features and classification targets; $\overline{r_{ff}}$ indicates the average correlation between features and features.

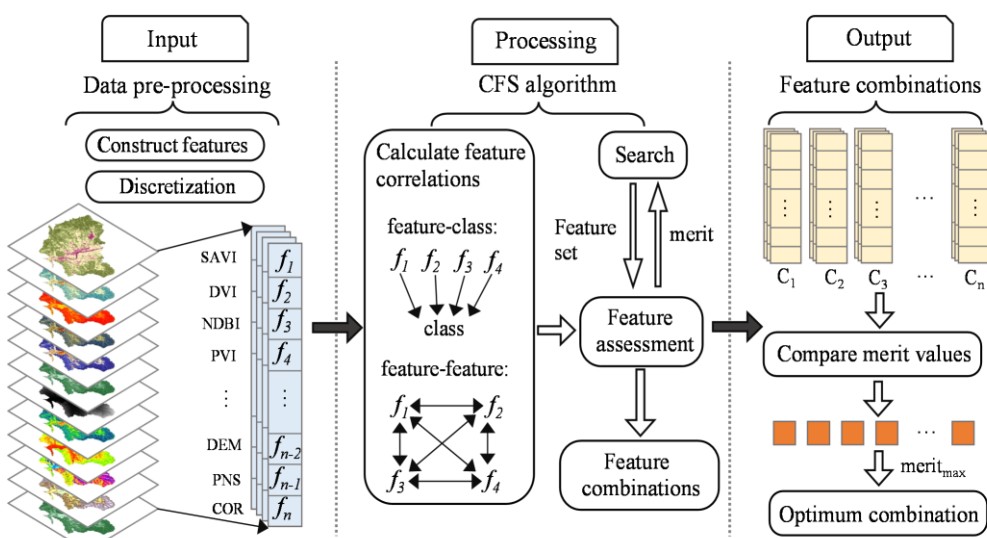

**Figure 4.** Feature selection and combination process based on the CFS algorithm.

2.3.3. Machine Learning Classification Algorithms

(A)　Random Forest

Random forest (RF) is a supervised machine learning classifier that contains multiple decision trees. In the process of building a decision tree, each tree sample is randomly extracted from the original data using the Bootstrap method and reorganised to form a subset as large as the original data. Moreover, each tree generates rules and classification results that match its properties. Eventually, the rules and classification results of all trees are integrated to achieve the classification. The main steps of the RF algorithm are as follows: (1) N training samples are extracted from the original sample set using the Bootstrap method with a put-back. A total of K rounds of extraction are performed to obtain K training sets (training sets are independent of each other); (2) one model is obtained using one training set at a time, and a total of K models are obtained for the K training sets; (3) the K models obtained in the previous step are used in a voting process to obtain the classification results.

(B)　CatBoost

CatBoost is a new open-source machine-learning algorithm proposed by Yandex in 2017. It consists of categorical features and gradient boosting and is a gradient-boosting decision tree (GBDT) framework with an oblivious tree as the base learner [44]. However, compared to the GBDT algorithm, CatBoost has made relatively significant improvements in handling category features, the Boosting approach, and the decision tree growth scoring. CatBoost integrates multiple base learners in a serial approach during model training. It ensures that the training sample set remains the same in each round, continuously updating the sample weights with the results of the previous round, thus gradually reducing the bias caused by noise points. There is a dependency relationship between the multiple weak learners generated by the training, and the final result is obtained by weighting the regression values of all the weak learners. In addition, it can efficiently and reasonably handle categorical features and deal with gradient bias and prediction shift problems, thus reducing the occurrence of overfitting and improving the generalisation ability of the algorithm.

(C)　ExtraTrees

Extremely randomised trees (ExtraTrees) is very similar to the RF algorithm, which is the integration model of decision trees [45]. However, the difference is that RF builds a decision tree by selecting an optimal feature value to divide the nodes based on principles such as information gain, Gini coefficient, and standard deviation, but ExtraTrees randomly

selects a feature and threshold to divide the tree. This increases the randomness and variability of each decision tree, which can sufficiently suppress the overfitting of the model and avoid bias by extreme samples.

### 2.3.4. Accuracy Assessment

When evaluating the classification accuracy of all land types, the overall accuracy (OA) and Kappa coefficient were chosen in this paper. When evaluating the accuracy of rice extraction, the F1 score was chosen.

OA is expressed as:

$$OA = \frac{\sum_{i=1}^{n} p_{i,i}}{N} \tag{5}$$

where $p_{i,i}$ represents the total number of correctly classified land type $i$, $n$ represents the total number of land types, and $N$ is the total number of samples.

Kappa is expressed as:

$$Kappa = \frac{N^2 * OA - \sum_{i=1}^{n} a_i b_i}{N^2 - \sum_{i=1}^{n} a_i b_i} \tag{6}$$

where $a_i$ is the number of actual samples in each land type, and $b_i$ is the number of predicted samples in each land type.

When evaluating the accuracy of rice extraction, this paper divided pixels into four types: correctly identified as rice, correctly identified as other classes, incorrectly identified as rice, and incorrectly identified as other classes, denoted by the true positive (TP), true negative (TN), false positive (FP), and false negative (FN), respectively. The F1 score is the harmonic average of precision and recall. Precision and recall are defined as:

$$Precision = \frac{TP}{TP + FP} \tag{7}$$

$$Recall = \frac{TP}{TP + FN} \tag{8}$$

The F1 score is expressed as:

$$F1 = \frac{2 Precision * Recall}{Precision + Recall} \tag{9}$$

## 3. Results

### 3.1. Analysis of the Spectral Time-Series Curves of Different Land Types

The study plotted spectral curves by counting the spectral averages of the land types in the different periods' images, as shown in Figure 5. Overall, the separation was high in the red edge-2 (b6) to the water vapour band (b9) for all land types, and the separation between vegetation and non-vegetation was most evident in the joining and heading periods. However, the spectral curves of vegetation exhibited different levels of overlap in different periods. Therefore, in order to effectively distinguish the different vegetation and extract rice, it was necessary to analyse the separability between vegetation further.

### 3.2. Optimal Extraction Periods of Features

One-way ANOVA requires that the data follow a normal distribution, but the data cannot achieve the desired state in most cases. If the absolute value of the kurtosis is less than 10 and the absolute value of the skewness is less than 3, the data can be accepted as having a normal distribution [46]. In this paper, the kurtosis and skewness of the features were calculated based on the different growth stages of rice and the different vegetation, as shown in Table 3. It can be seen that the kurtosis and skewness of all features satisfied the condition, indicating that the data largely had a normal distribution. In addition, the average and standard deviation of the features were calculated using vegetation samples,

and the *p*-values of all features were less than 0.01. This shows that the different vegetation types differed significantly in each feature treatment.

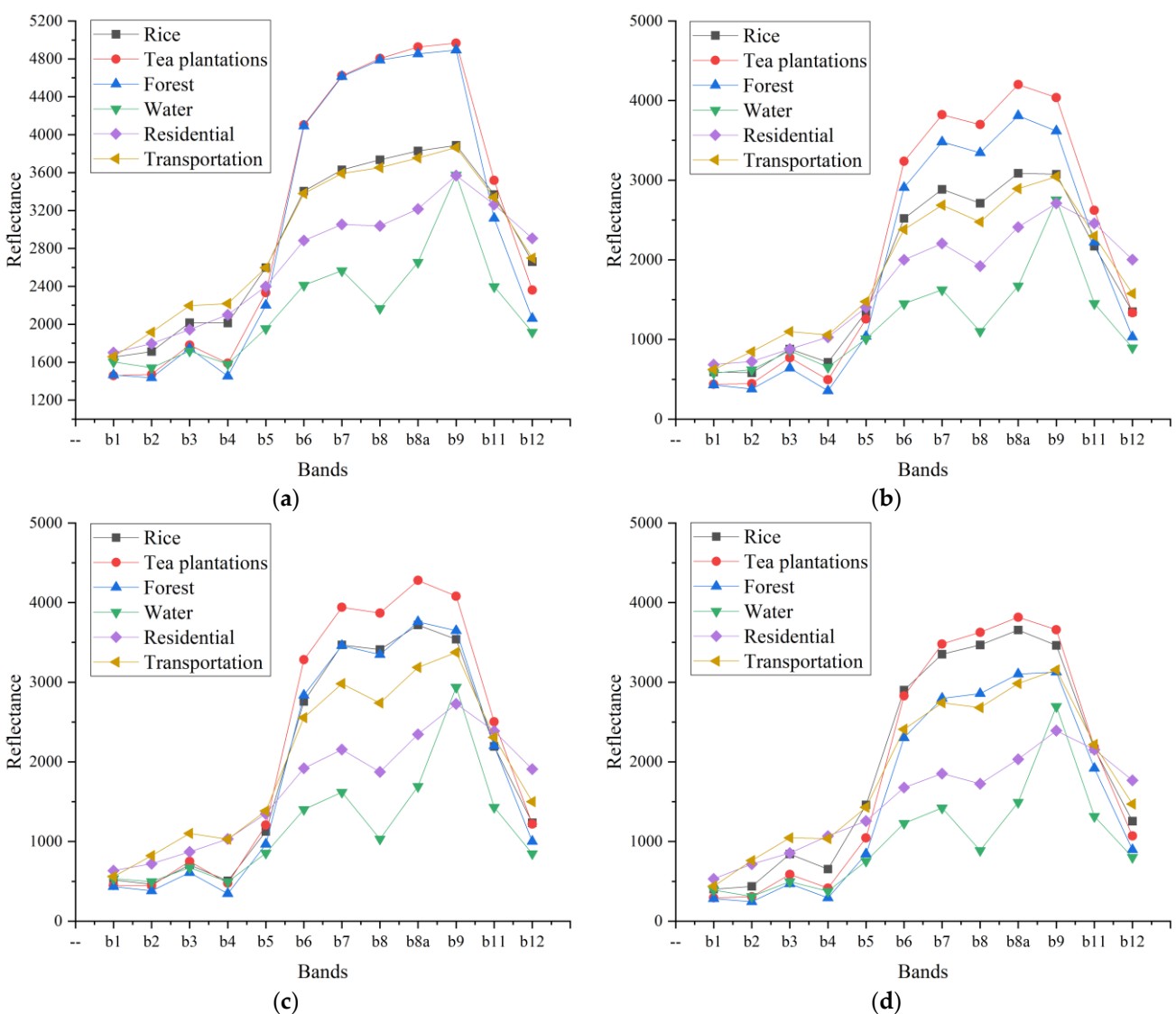

**Figure 5.** Spectral curves of land types in different growth stages of rice. (**a**) Seeding period, (**b**) joining period, (**c**) heading period, and (**d**) maturity period.

**Table 3.** Kurtosis and skewness of features in different periods.

|  | **PNS** | | | | **SAVI** | | | | **RVI** | | | |
|---|---|---|---|---|---|---|---|---|---|---|---|---|
|  | May | Jun | Jul | Sept | May | Jun | Jul | Sept | May | Jun | Jul | Sept |
| **Kurtosis** | −2.45 | −2.09 | −1.68 | −1.73 | −0.20 | −0.73 | −1.55 | −1.18 | 0.48 | 0.42 | −0.50 | 0.44 |
| **Skewness** | 1.82 | 2.64 | 0.88 | 0.98 | −1.04 | −0.34 | 2.29 | 1.55 | −0.83 | −1.01 | −0.33 | −0.63 |
|  | **NDWI** | | | | **NDVI$_{re3}$** | | | | **NDVI** | | | |
|  | May | Jun | Jul | Sept | May | Jun | Jul | Sept | May | Jun | Jul | Sept |
| **Kurtosis** | 0.14 | 0.60 | 1.06 | 0.80 | 0.17 | −0.25 | −0.29 | −0.07 | −0.20 | −0.73 | −1.55 | −1.18 |
| **Skewness** | −0.86 | −0.6 | 1.16 | 0.34 | 0.55 | 0.47 | 0.15 | 0.30 | −1.04 | −0.34 | 2.29 | 1.55 |
|  | **NDBI** | | | | **LSWI** | | | | **EVI** | | | |
|  | May | Jun | Jul | Sept | May | Jun | Jul | Sept | May | Jun | Jul | Sept |
| **Kurtosis** | −0.08 | 0.41 | 0.45 | 0.56 | 0.09 | −0.52 | −0.43 | −0.41 | −0.06 | −0.67 | −1.03 | −0.81 |
| **Skewness** | −1.01 | −0.33 | 0.39 | 0.18 | −0.97 | −0.42 | 0.45 | 0.10 | −1.07 | −0.01 | 1.39 | 0.40 |

**Table 3.** *Cont.*

| | DVI | | | | CARI | | | | PVI | | | |
|---|---|---|---|---|---|---|---|---|---|---|---|---|
| | May | Jun | Jul | Sept | May | Jun | Jul | Sept | May | Jun | Jul | Sept |
| **Kurtosis** | 0.33 | 0.13 | −0.22 | −0.26 | 0.03 | 0.18 | 0.34 | −0.42 | 0.33 | 0.13 | −0.21 | −0.25 |
| **Skewness** | −0.82 | −0.76 | 0.06 | 0.00 | −0.71 | −0.89 | −0.14 | 0.22 | −0.81 | −0.75 | 0.05 | −0.03 |
| | **Mean** | | | | **HOM** | | | | **ENT** | | | |
| | May | Jun | Jul | Sept | May | Jun | Jul | Sept | May | Jun | Jul | Sept |
| **Kurtosis** | −0.30 | −0.22 | −0.13 | 0.47 | −0.1 | 0.25 | 0.05 | 0.18 | −0.77 | −0.89 | −0.79 | −0.93 |
| **Skewness** | 0.27 | −0.74 | 0.16 | 0.17 | −0.36 | −0.15 | −0.35 | −0.26 | 0.17 | 0.41 | 0.27 | 0.30 |
| | **COR** | | | | **PCA1** | | | | **PCA2** | | | |
| | May | Jun | Jul | Sept | May | Jun | Jul | Sept | May | Jun | Jul | Sept |
| **Kurtosis** | −0.78 | −0.73 | −0.80 | −0.81 | −0.45 | −0.17 | −0.16 | 0.49 | 0.16 | −0.40 | 1.64 | −1.31 |
| **Skewness** | −0.18 | −0.43 | −0.17 | −0.31 | 0.01 | −0.77 | 0.17 | 0.15 | −0.92 | −0.35 | 5.77 | 2.91 |

The magnitude of vegetation difference was calculated based on the partial $\eta^2$ and Cohen's f indicators, and the quantified results are shown in Table 4 (excluding terrain features). Taking the PNS index as an example, among the different growth stages of rice, the partial $\eta^2$ and Cohen's f were the largest in July (heading period), which indicated that the PNS index made the highest separability between rice, tea plantations, and forest in the heading period. Similarly, the 13 features of AVI, RVI, NDWI, NDVI, NDBI, LSWI, EVI, DVI, PVI, Mean, ENT, PCA1, and PCA2 for the vegetation had the highest separability in May (seeding period). In addition, the four features of NDVI$_{re3}$, CARI, HOM, and COR for the vegetation had the highest separability in June (joining period). The optimal temporal features were mainly concentrated in the seeding and growth periods of rice.

**Table 4.** Quantitative results of vegetation separability under different period features.

| | PNS | | | | SAVI | | | | RVI | | | |
|---|---|---|---|---|---|---|---|---|---|---|---|---|
| | May | Jun | Jul | Sept | May | Jun | Jul | Sept | May | Jun | Jul | Sept |
| **Partial $\eta^2$** | 0.06 | 0.15 | 0.18 | 0.08 | 0.67 | 0.57 | 0.12 | 0.31 | 0.66 | 0.60 | 0.13 | 0.10 |
| **Cohen's f** | 0.25 | 0.42 | 0.46 | 0.29 | 1.42 | 1.14 | 0.36 | 0.67 | 1.40 | 1.23 | 0.38 | 0.33 |
| | **NDWI** | | | | **NDVIre3** | | | | **NDVI** | | | |
| | May | Jun | Jul | Sept | May | Jun | Jul | Sept | May | Jun | Jul | Sept |
| **Partial $\eta^2$** | 0.70 | 0.57 | 0.05 | 0.45 | 0.03 | 0.12 | 0.06 | 0.05 | 0.67 | 0.57 | 0.12 | 0.31 |
| **Cohen's f** | 1.52 | 1.19 | 0.23 | 0.90 | 0.19 | 0.37 | 0.25 | 0.24 | 1.42 | 1.14 | 0.36 | 0.67 |
| | **NDBI** | | | | **LSWI** | | | | **EVI** | | | |
| | May | Jun | Jul | Sept | May | Jun | Jul | Sept | May | Jun | Jul | Sept |
| **Partial $\eta^2$** | 0.50 | 0.19 | 0.01 | 0.05 | 0.55 | 0.24 | 0.00 | 0.03 | 0.63 | 0.39 | 0.17 | 0.35 |
| **Cohen's f** | 0.99 | 0.48 | 0.08 | 0.23 | 1.11 | 0.56 | 0.03 | 0.17 | 1.31 | 0.79 | 0.45 | 0.74 |
| | **DVI** | | | | **CARI** | | | | **PVI** | | | |
| | May | Jun | Jul | Sept | May | Jun | Jul | Sept | May | Jun | Jul | Sept |
| **Partial $\eta^2$** | 0.61 | 0.48 | 0.10 | 0.11 | 0.46 | 0.53 | 0.11 | 0.12 | 0.61 | 0.48 | 0.10 | 0.11 |
| **Cohen's f** | 1.26 | 0.96 | 0.33 | 0.34 | 0.93 | 1.06 | 0.35 | 0.40 | 1.26 | 0.95 | 0.34 | 0.35 |
| | **Mean** | | | | **HOM** | | | | **ENT** | | | |
| | May | Jun | Jul | Sept | May | Jun | Jul | Sept | May | Jun | Jul | Sept |
| **Partial $\eta^2$** | 0.46 | 0.45 | 0.24 | 0.26 | 0.03 | 0.03 | 0.01 | 0.01 | 0.02 | 0.02 | 0.00 | 0.01 |
| **Cohen's f** | 0.93 | 0.90 | 0.56 | 0.59 | 0.17 | 0.16 | 0.10 | 0.09 | 0.15 | 0.12 | 0.06 | 0.12 |
| | **COR** | | | | **PCA1** | | | | **PCA2** | | | |
| | May | Jun | Jul | Sept | May | Jun | Jul | Sept | May | Jun | Jul | Sept |
| **Partial $\eta^2$** | 0.00 | 0.01 | 0.01 | 0.01 | 0.43 | 0.42 | 0.21 | 0.26 | 0.68 | 0.47 | 0.07 | 0.21 |
| **Cohen's f** | 0.06 | 0.12 | 0.11 | 0.09 | 0.88 | 0.85 | 0.52 | 0.59 | 1.46 | 0.94 | 0.28 | 0.51 |

### 3.3. Results of Feature Selection

Table 5 lists the feature-selection results for each period (SC-Seeding, SC-Joining, SC-Heading, SC-Maturity) and the result of multi-period feature selection (MC) based on OPFSM. It can be seen that the DEM, Slope, and PNS indices appeared in all feature combinations. This indicates the high stability of terrain and spectral-spatial features in the feature selection.

**Table 5.** Preferred results for features.

| Combinations | Preferred Features |
|---|---|
| SC-Seeding | DEM, Slope, PNS, NDWI, EVI, Mean, SAVI |
| SC-Joining | DEM, Slope, PNS, PVI, NDVIre3, NDBI, CARI, Mean, SAVI |
| SC-Heading | DEM, Slope, PNS, NDWI, NDVIre3, NDBI, EVI, DVI, Mean, Curvature |
| SC-Maturity | DEM, Slope, PNS, NDWI, NDVIre3, NDBI, PCA1, CARI |
| Multitemporal | DEM, Slope, $PNS_{Jul}$, $NDWI_{May}$, $NDVI_{re3Jun}$, $ENT_{May}$, $EVI_{May}$, $CARI_{Jun}$, $Mean_{May}$, $SAVI_{May}$ |

### 3.4. Accuracy Assessment and Classification Results

The original remote sensing images (ORIGs), SCs, and MC were classified using RF, CatBoost, and ET classifiers, respectively. Nine classification schemes could be formed: ORIGs-RF, SCs-RF, MC-RF, ORIGs-CatBoost, SCs-CatBoost, MC-CatBoost, ORIGs-ET, SCs-ET, and MC-ET. The OA and Kappa of each classification scheme based on different periods are shown in Figure 6. Overall, the classification accuracy from high to low was MC > SCs > ORIGs. In ORIGs, the relative classification accuracy in July was the lowest. However, the accuracy in this period became significantly higher in SCs. The result illustrates the improvement of spectral confusion in the dense vegetation period through feature extraction. In MC, the combination based on the ET classifier (MC-ET) had the highest accuracy, with the OA of 86% and Kappa of 0.81. The classification accuracy in the MC was ranked from high to low as MC-ET > MC-CatBoost > MC-RF.

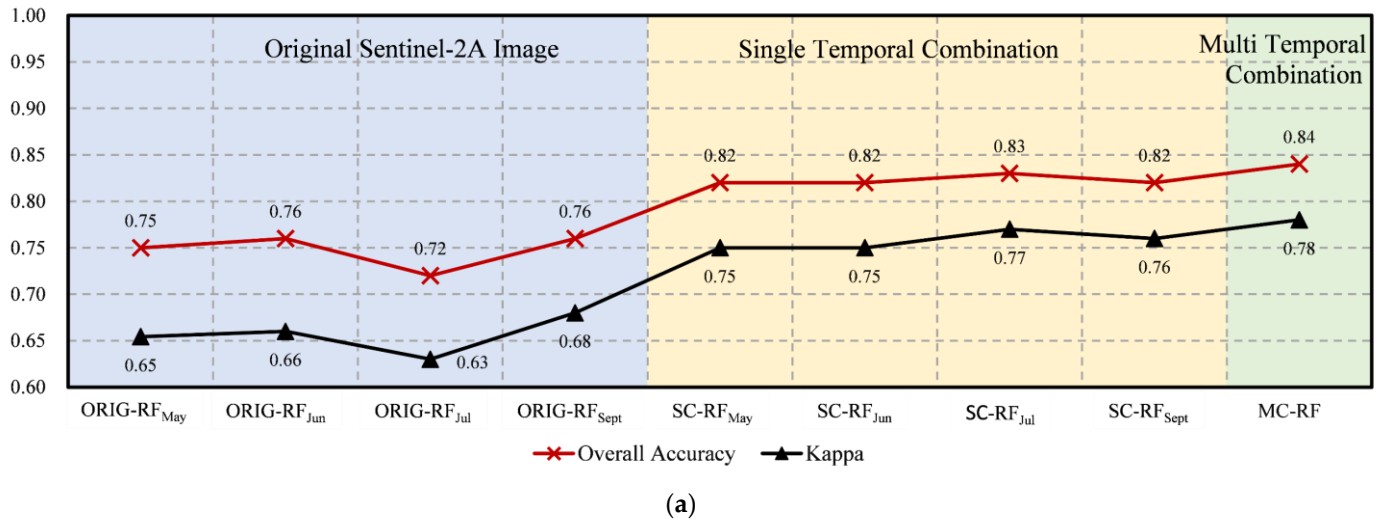

(**a**)

**Figure 6.** *Cont.*

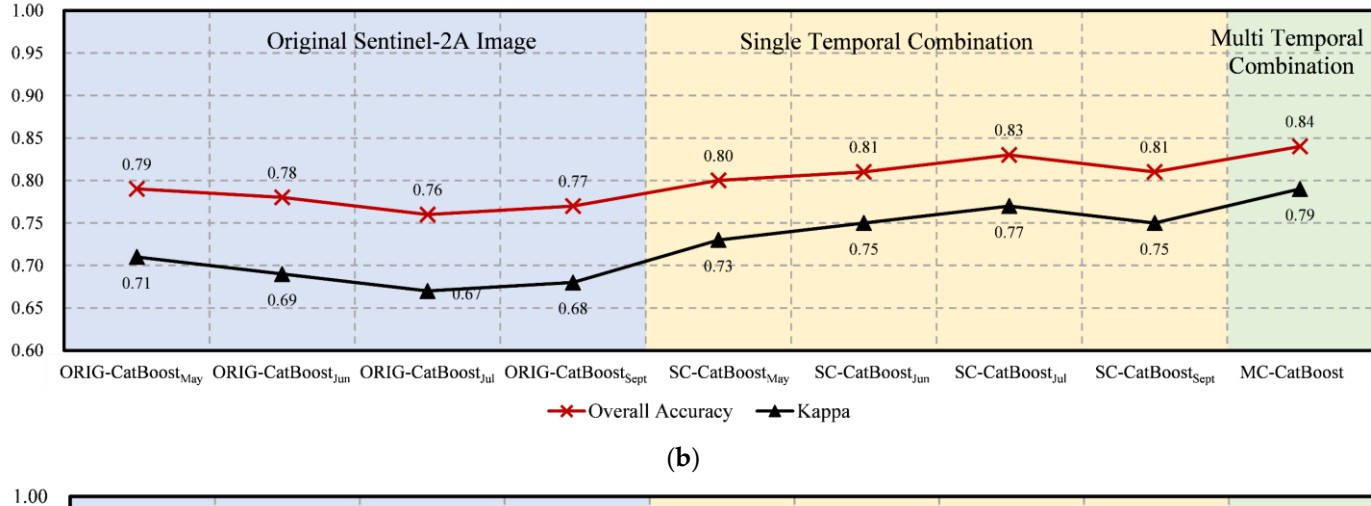

(**b**)

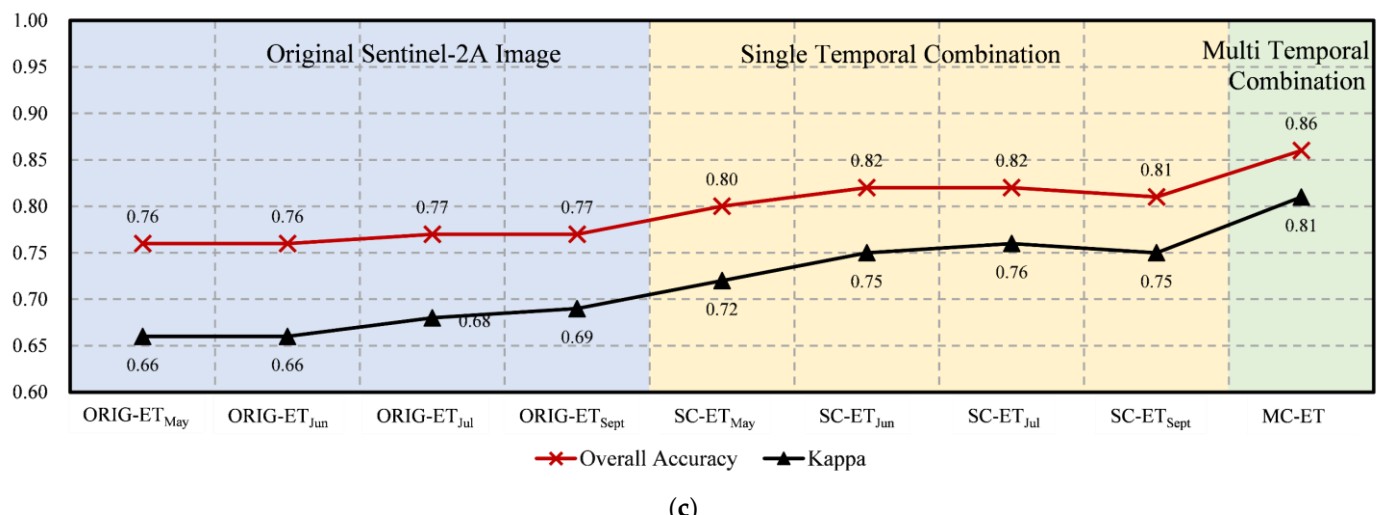

(**c**)

**Figure 6.** OA and kappa of different classification schemes. Classification accuracy based on (**a**) RF, (**b**) CatBoost, and (**c**) ET algorithms.

The F1 scores for rice based on the different classification schemes are shown in Table 6. In the ORIGs-based classification schemes (ORIGs-RF, ORIGs-CatBoost, ORIGs-ET), the F1 score of rice was highest in May (seeding period) and lowest in July (heading period), and the performance of each classifier was relatively stable. In the SCs-based classification schemes (SCs-RF, SCs-CatBoost, SCs-ET), the F1 scores for different growth stages of rice were not significantly different, but the accuracy improvement was still the greatest in the heading period. In the MC-based classification schemes (MC-RF, MC-CatBoost, MC-ET), the accuracy was ranked from high to low as MC-ET > MC-CatBoost > MC-RF. The F1 of MC-ET was 0.95, so the MC-ET can be considered the best classification scheme for rice extraction.

**Table 6.** F1 scores for rice under different classification schemes.

| | ORIG-RF | | | | SC-RF | | | | MC-RF |
|---|---|---|---|---|---|---|---|---|---|
| | May | Jun | Jul | Sept | May | Jun | Jul | Sept | |
| **F1 Score** | 0.86 | 0.83 | 0.77 | 0.84 | 0.90 | 0.88 | 0.91 | 0.91 | 0.93 |
| | ORIG-CatBoost | | | | SC-CatBoost | | | | MC-CatBoost |
| | May | Jun | Jul | Sept | May | Jun | Jul | Sept | |
| **F1 Score** | 0.87 | 0.87 | 0.80 | 0.85 | 0.89 | 0.88 | 0.90 | 0.91 | 0.93 |

**Table 6.** *Cont.*

| | ORIG-ET | | | | SC-ET | | | | MC-ET |
|---|---|---|---|---|---|---|---|---|---|
| | May | Jun | Jul | Sept | May | Jun | Jul | Sept | |
| **F1 Score** | 0.86 | 0.84 | 0.82 | 0.85 | 0.90 | 0.89 | 0.90 | 0.91 | 0.95 |

Based on the results of the preceding analysis, it can be concluded that MC stretched the attribute differences between vegetation and reflected the extraction period of features. By comparison, MC-ET achieved the highest accuracy in rice extraction. Therefore, MC-ET was used to extract and map mountain rice. As shown in Figure 7a, rice was mainly concentrated in patches along rivers and roads and around residential areas, along with some in strips along mountain valleys. Plus, it had an excellent distinction at the junction of rice and other land types. However, the "misclassification validation" samples for all land types were primarily distributed along roads and in fragmented areas of different land types (Figure 7b). The resolution of the imagery may represent the cause of the error.

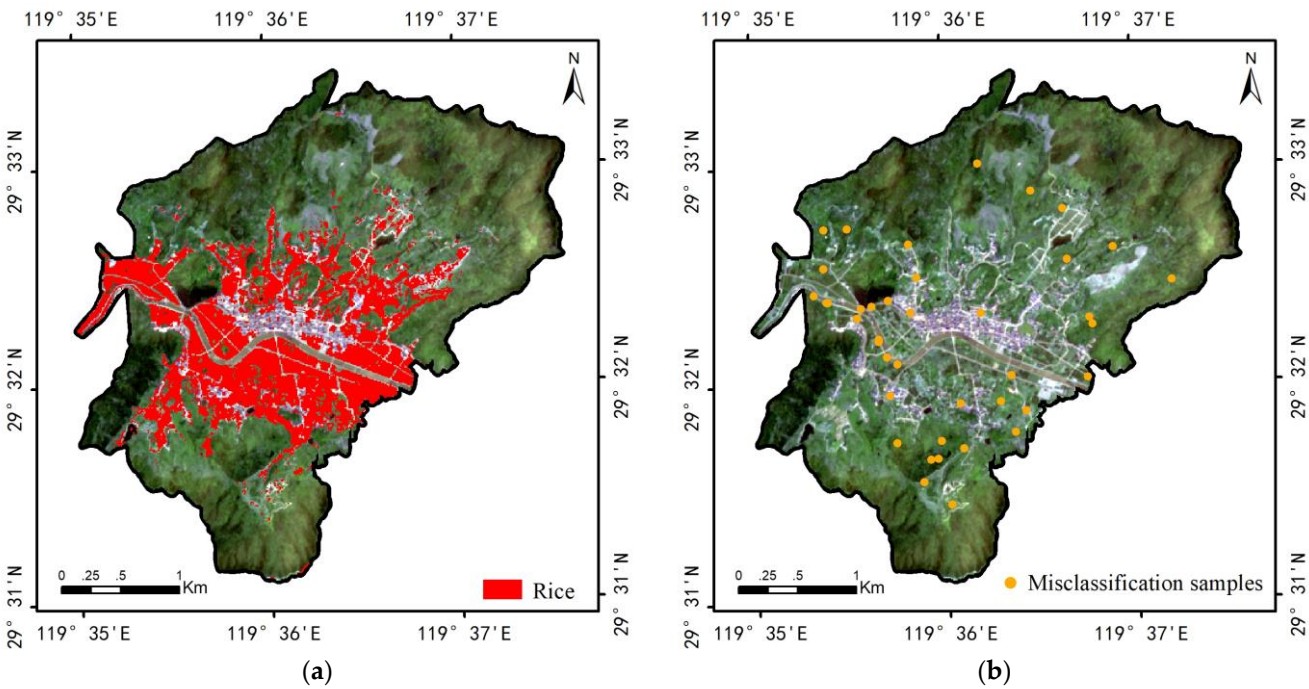

**Figure 7.** Map of the optimal distribution of mountain rice and the distribution of "misclassification verification" samples. (**a**) Optimal distribution map of mountain rice, (**b**) distribution map of misclassification verification samples.

## 4. Discussion

### 4.1. Feature Analysis

Tu et al. [47] noted that rich features could effectively reflect the differences between land types and distinguish their boundaries to improve the accuracy of image classification. This study ensured the diversity of features when compared to the extraction of land types using spectral and phenological information alone. Among them, it should be emphasised that the spectral-spatial feature has been used less in mountain crop extraction. However, it has been shown that using spatial-spectral information can improve classification accuracy, more so than using spectral features or spatial features alone [48–50]. In particular, for the boundaries of different land types, the spectral-spatial feature can balance spectral and spatial information's heterogeneity. In recent studies, many methods for extracting the spectral-spatial feature have been proposed [51–53]. One of the relatively simple methods is vector stacking, where different features are combined into long vectors. For example,

spatial information was directly stacked into spectral features using filters such as 3D wavelets [54] or 3D Gabor wavelets [55]. However, overlay strategies not only produced higher-dimensional features but also treated all features equally, making it difficult to explain the feature that played a vital role in the classification. In this study, the PNS index was used to determine the spectral matching by calculating the cosine similarity between neighbouring pixels, an approach that connected the spectral information from all channels simultaneously and considered the spatial relationship and the distance between pixels. As mountainous fields are relatively fragmented, the way of constructing the PNS index may provide some reference for improving the distinction of boundary land types. In addition, the features were manually extracted in this study. In recent years, deep learning has been considered a breakthrough technology in the field of machine learning and data mining, including the field of remote sensing [56,57]. By repeatedly training the input data in an artificial neural network, the features for a given classification task can be extracted automatically. This process does not require the processing of predefined features and can objectively solve problems such as classification and regression [58,59]. However, the training model requires a large amount of data and powerful hardware and software equipment. The accuracy of its performance is often not ideal when the data volume is small. Moreover, the model has the problem of being unexplainable compared to manual methods. However, traditional and deep learning methods are complementary, and we will actively explore deep learning methods in the future.

### 4.2. Analysis of Feature Selection Using OPFSM

For feature selection of multitemporal imagery, Song et al. [60] involved all features in the feature-selection algorithm simultaneously. Alternatively, only the important features were selected to extract land types [30]. Although these studies perform better in classification accuracy, the same feature may have similar performance and quality in different periods, leading to a feature being selected repeatedly or never. To avoid this problem, the OPFSM constructed in this study only placed the features of the optimal extraction period into the CFS algorithm for screening. We referred to the approaches used in traditional studies, i.e., where the features from all periods were directly involved in the selection, and we treated the selection result as a comparison scheme, i.e., Scheme 1. The top 10 features with high importance (the number of features remained the same as MC) were selected only based on the OOB error in RF and the result was also used as a comparison scheme, i.e., Scheme 2. We compared the optimal results (MC) of OPFSM with Schemes 1 and 2. As shown in Table 7, it is easy to see that a total of 20 features were selected from the 72 initial features (four periods) in Scheme 1. NDBI, CARI, PNS, NDWI, NDVI, SAVI, and EVI were repeated two times in Scheme 1. NDWI occurred three times in Scheme 2. Therefore, it is difficult to guarantee the heterogeneity of features. In contrast, MC contained various features, and there were no repetitions.

**Table 7.** Comparison of different feature-selection schemes.

| Comparison Schemes | Features |
|---|---|
| Scheme 1 | DEM, Slope, Curvature, $NDBI_{Jul}$, $NDBI_{Sept}$, $CARI_{Jun}$, $CARI_{Sept}$, $PNS_{Jun}$, $PNS_{Sept}$, $NDWI_{May}$, $NDWI_{Sept}$, $NDVI_{re3Jun}$, $NDVI_{re3Jul}$, $SAVI_{May}$, $SAVI_{Jun}$, $EVI_{May}$, $EVI_{Jul}$, $Mean_{Jun}$, $DVI_{Jul}$, $PCA_{Sept}$ |
| Scheme 2 | DEM, Slope, $NDWI_{May}$, $NDWI_{Sept}$, $NDWI_{Jul}$, $NDBI_{Jul}$, $CARI_{Jun}$, $EVI_{Jul}$, $DVI_{Jul}$, $PCA_{Sept}$ |
| MC | DEM, Slope, PNSJul, $NDWI_{May}$, $NDVI_{re3Jun}$, $ENT_{May}$, $EVI_{May}$, $CARI_{Jun}$, $Mean_{May}$, $SAVI_{May}$ |

Similarly, Zhu et al. [32] emphasised that pursuing feature separability or importance alone is undesirable in feature selection. This conclusion was also verified when the F1 scores of rice in this study were compared (Figure 8). It can be seen that MC had the highest F1 score in the ET algorithm. In addition, when compared to Scheme 2, the MC had a relatively high F1 score when we were using the RF algorithm. Although some F1 scores were relatively flat at 0.93, the number of features in Scheme 1 was excessive, and the lack of

texture and spectral-space features in Scheme 2 did not ensure feature diversity. In general, the F1 scores were relatively lower in Schemes 1 and 2 than in MC.

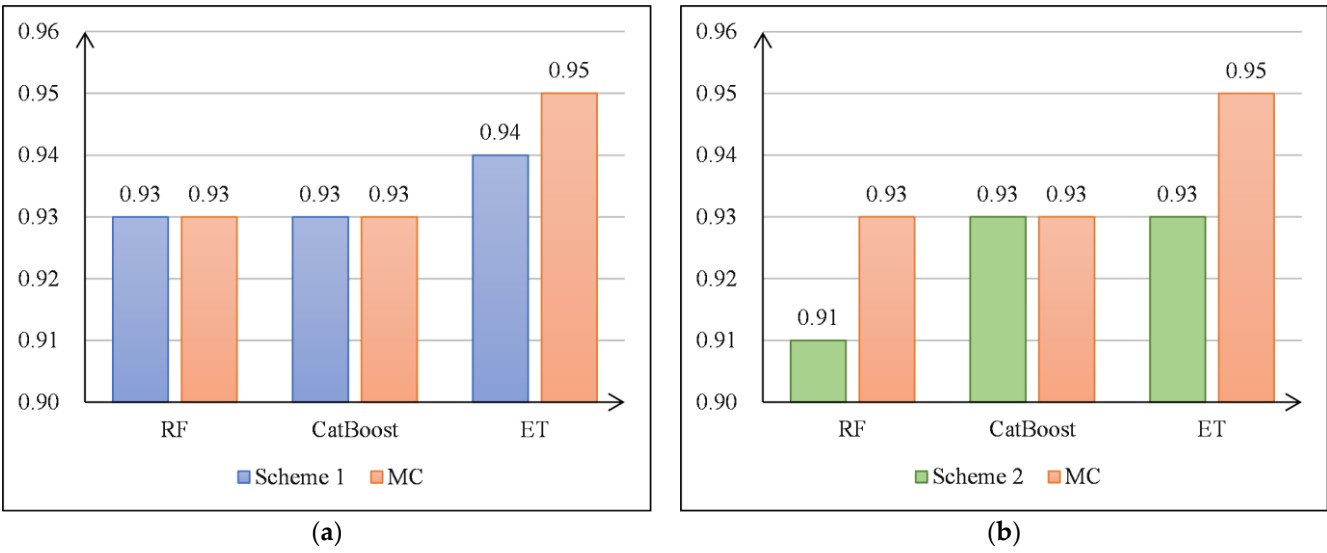

(**a**)    (**b**)

**Figure 8.** F1 scores for rice under different comparison schemes. (**a**) MC compared with Scheme 1, (**b**) MC compared with Scheme 2.

In addition, when analysing the correlation between features, the initial features were permuted and combined to form 231 feature pairs (excluding themselves), which contained a total of 40 feature pairs with very high correlation (Pearson's correlation coefficient $\geq 0.8$ or $\leq -0.8$) (Figure 9a). However, the 45 feature pairs in MC contained only three feature pairs with high correlation (Figure 9b). The highly correlated feature pairs decreased from 17.32% to 6.67%. This shows that the MC has low redundancy, and OPFSM can provide a new reference for the feature selection of multitemporal images.

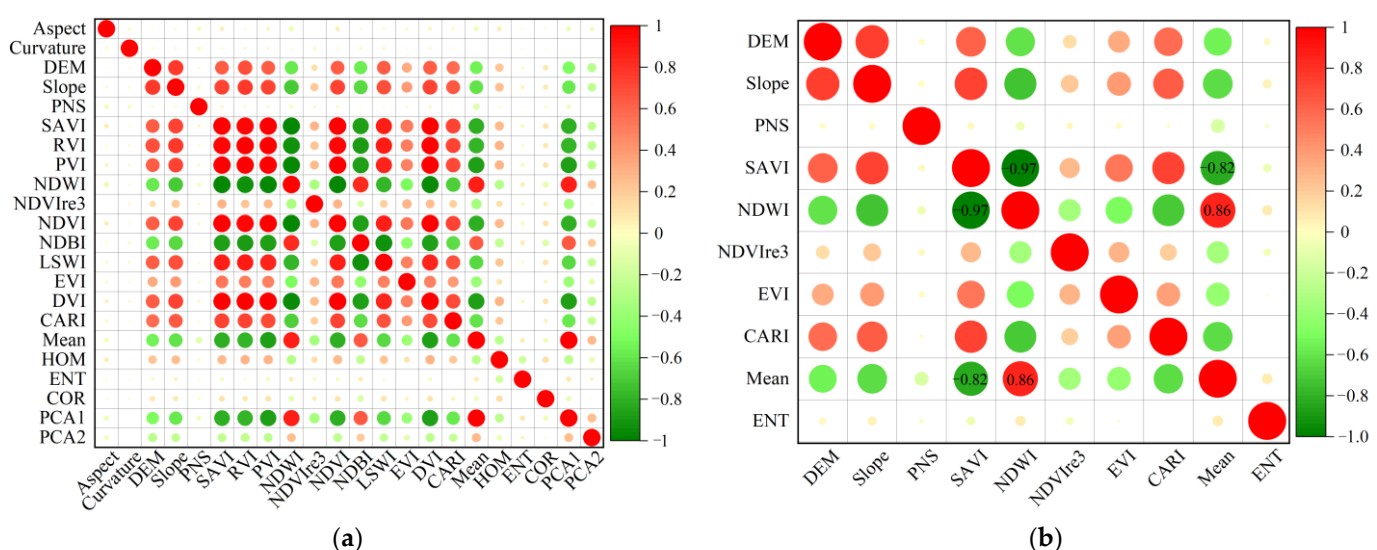

(**a**)    (**b**)

**Figure 9.** Comparison of correlation between initial features and between preferred features. (**a**) Correlation between initial features, (**b**) correlation between features in MC.

### 4.3. Performance of Machine Learning Classifiers when Extracting Mountain Rice

RF, CatBoost, and ET are typical ensemble machine-learning algorithms [61,62]. Compared with RF and ET, CatBoost uses a serial way to construct decision trees, which constantly updates the sample weights from the previous round of results to improve the

training of the next tree. CatBoost has a good generalisation capability, but it performs poorly in conditions of abnormal and limited samples [31]. In addition, RF and ET use a parallel approach to construct decision trees, and each tree is constructed using independent samples from the training set. Limited and abnormal samples have fewer influence results. However, each decision tree in ET has greater randomness and variability and is less affected by abnormal samples [63]. As shown in Figure 6 and Table 6, ET tended to outperform RF and CatBoost in extracting land types. Therefore, ET may have been more suitable for mountain rice extraction in this study.

However, different classifiers are adapted to different application scenarios. Research should not be limited to a single algorithm, to ensure the stability and robustness of the results. The deep neural network algorithm (DNN) has also been widely used for classification issues [64]. However, the demand for the large sample size of DNN may affect how it can be applied. In this study, the sample sizes in the hundreds did not satisfy the creiteria of DNN.

### 4.4. Variability of the Rice Growth Cycle

The best periods for features were the early growth periods of rice in this study, i.e., the seeding and growth periods. However, different regions may have different rice varieties, soils, and climates, which result in inconsistent rice growth cycles [65]. Limiting the study to a specific period alone does not make all results reliable, but rice exhibits similar characteristics in different growth periods. Therefore, this study mainly explored how to fully integrate the temporal and feature information of crops, i.e., how to reflect the multitemporal advantages while maintaining the diversity and heterogeneity of features in the preferred results.

### 5. Conclusions

This study constructed a feature-selection method based on the optimal extraction period of each feature (OPFSM) to extract mountain rice. A multitemporal feature combination (MC) was formed using OPFSM. The rice extraction accuracy of MC was also compared with single-temporal feature combinations (SCs) and original remote sensing images (ORIGs) using Random Forest, CatBoost, and ET algorithms. The results of our study allowed for three conclusions to be inferred. First, among all the features, only DEM, Slope, and PNS indices were stable in all feature combinations (MC, SCs, OIRGs) after feature selection. This indicated that terrain and spectral-spatial features have great potential for the extraction of mountain rice. Second, when comparing the correlations between features, MC reduced the high-correlation feature pairs by 12.38% compared with all the features, with lower redundancy. Finally, the classification accuracy of MC was relatively stable, and was highest under different classifiers (MC > SCs > ORIGs). The MC-ET classification scheme had the highest accuracy (OA 86%, Kappa 0.81, F1 score of rice 0.95). Overall, our results have the potential to improve the extraction accuracy of mountain rice.

These results demonstrated a timely, simple, and cost-effective framework for assessing mountain rice with publicly available data, which could contribute to enhancing agricultural production and management. Future investigations could explore the potential for crop extraction by combining SAR and optical imagery, as well as the ability of deep learning to advance the results of this study and expand the scope of the study by harnessing the powerful cloud computing capabilities of GEE (GoogleEarth Engine).

**Author Contributions:** Conceptualisation, K.Z. and Y.C.; methodology, K.Z.; software, W.W.; validation, J.H., W.W. and K.Z.; formal analysis, K.Z.; investigation, B.Z.; resources, B.Z. and J.H.; data curation, B.Z.; writing—original draft preparation, K.Z.; writing—review and editing, Y.C.; visualisation, W.W.; supervision, J.H. and B.Z.; project administration, K.Z. and Y.C.; funding acquisition, Y.C. All authors have read and agreed to the published version of the manuscript.

**Funding:** This work was supported by the National Natural Science Foundation of China (Grant No. 41201408) and the Zhejiang Province Natural Science Foundation of China (Grant No. LY16D010009).

**Data Availability Statement:** Not applicable.

**Acknowledgments:** The authors would like to thank the editors and reviewers for their suggestions and revisions.

**Conflicts of Interest:** The authors declare no conflict of interest.

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
