# Peer review of "A Multitemporal Mountain Rice Identification and Extraction Method Based on the Optimal Feature Combination and Machine Learning"

_remotesensing, doi:10.3390/rs14205096_

Round 1

Reviewer 1 Report

Dear Authors,

The authors present paper A Multi-Temporal Mountain Rice Identification and Extraction Method Based on Optimal Feature Combination and Machine Learning.The manuscript is well-written and nicely presented, as well.
The Introduction: Provides a clear idea of extracting information from crops based on remote sensing technology
Materials and methods describe the study area. A map of study area is also included. Next,  Data, Data Image and Digital Terrain Data.
 Methods: The methods used are very well explained in sub-sections: Feature Extraction, Feature Selection and Combination, Machine Learning Classification Algorithms, Accuracy assessment.
The results are very amply and detailed presented and also with tables, graphs and figures in suub-sections: Analysis of the Spectral Time Series Curves of Different Land Type, Quantitative Analysis of Feature Differences of Different Vegetation, Feature Combination Results and Correlation Analysis between Features, Accuracy Assessment and Classification Results
 Discussions are too limited for such results. So, this section needs to be developed.
The conclusions reflect the objectives very briefly but well.

Author Response

Q1、English language and style are fine/minor spell check required.

Response: We are grateful for the suggestion.We are sorry for the language problems in the original draft. We have improved the English writing in the whole paper.

Q2、Discussions are too limited for such results. So, this section needs to be developed.

Response: Thank you very much for your suggestion. It has changed my perception of the "Discussion" section ( I always thought of "Discussion" and "Conclusion" as similar until now). Thank you for your guidance!
I have rewritten the whole discussion. It mainly discusses feature selection, feature selection methods, classification algorithms, and so on.
Please refer to the attachment (annotation: Q2 of reviewer 1 in P19) for details.

Reviewer 2 Report

The authors proposed a multi-temporal mountain rice area detection method by applying classifiers such as RF and ET to 21 hand-crafted features. Satellite images and Dem etc. data of Sandu Town are used for experiments. The research topic is meaningful but the methodology is lack of novelty.  It is not necessary to only use hand-crafted features for detection , especially detection from images.  So I recommend not accept.

1. There are lots of mature deep learning-based methods such as unet. Why not use deep learning methods? Some data such as DEM might be a useful datasource, and is hard to be considered in deep learning methods. I think combination of deep learning and hand-crafted feature would be interesting and useful.

2. The whole manuscript should be further polished in English grammars and narrative logic.

Author Response

Q1: There are lots of mature deep learning-based methods such as unet. Why not use deep learning methods? Some data such as DEM might be a useful datasource, and is hard to be considered in deep learning methods. I think combination of deep learning and hand-crafted feature would be interesting and useful.

Response: Thank you very much for bringing up our shortcomings. In response to your suggestion, we tried to use U-net to extract features automatically, but since we knew less about this piece before, it took a lot of time to try. But unfortunately, we did not achieve the desired results in a short time. Considering that other contents need to be modified, we have to put this section in the "Discussion" section. We mainly discussed the advantages and disadvantages of using deep learning methods. Thank you again for your proposal, which is very important to us. Although the results are regrettable, they inspire us to explore further.

Please see the attachment for specific modifications. (P20, Comment in discussion: Q1 of reviewer 2).

In addition, in the last paragraph of the "Introduction", we describe the innovation of the article. In addition, we have made the following changes to the "Methodology": the original section 2.3.2 and 2.3.2 have been merged into "2.3.2. Feature selection" under which (A and B) are discussed separately. It is also referred to as "based on the optimal extraction period of each feature (OPFSM)". For details, please refer to the annex (P9, annotation: OPFSM)

Reviewer 3 Report

The paper is interesting. It is well constructed but two points need to be reworked: (1) the introduction as suggested in my comments (see pdf document) and (2) a better description of the originality of the study performed.

Author Response

Q1: The entire introduction is to be redone in its entirety because the key issues are not discussed:

- why Sentinel-2 and not Sentinel-1. For me it is more relevant to work with S1 and not with Sentinel-2 because there are often clouds

- discuss possible limitations due to the use of optical images: clouds etc. Discuss the literature that evokes this point, the number of optical images necessary, the period of S2 acquisitions: it is imperative to have images in which month/period/...

- Discuss the limitations related to the mountain (relief), the size of the plots and the adaptation or not of the S2 resolution

- Finally in the last paragraph I do not see any innovation in what you wrote: what is the new and the interesting aspect compared to other studies?

Response: Thank you for your suggestion to us. We have made the following changes in the introduction: (1) Data description. It mainly compares the advantages and disadvantages of radar images and optical images, the application range and the advantages of multi-real-phase images. (2) Feature selection. The diversity of features is ensured. (3) Feature selection method. The heterogeneity of multi-real-phase features is ensured. Thank you very much for your advice on "quotes", I have learned a lot in just a few days.

Please see the annex for detailed changes (P6)

Q2: Talk about the size of the plots. Talk about weather conditions: cloudiness

Response: Thank you for your suggestions, we have included them in "Study Area" and "2.2.1. Image Data"

Please see the annex for detailed changes (P4, annotation: study area size. P5 and P6, annotation: cloudiness)

In addition to that:

1、The “optimal temporal for each feature" in the abstract has been changed to "the optimal period for feature extraction".(P1,Annotation: descriptive modification)

2、Added "Sentinel-2" to "Keywords".(P1,Annotation: Add)

3、Title “2.2.2. Sample Data” was changed to “2.2.2. Ground Survey Data and Sample Datasets”(P3,Annotation: Title modification)

4、In "2.2.1. Image Data", “The main crops grown on a large scale in the study area are rice, tea plantations and citrus. However, with the development of modern agricultural technology and the demand for fruit in all seasons in recent years, greenhouse cultivation and late-maturing cultivation on trees are mainly used to prolong the fruiting period of citrus.” has been changed to : The crops are rice, tea plantations and citrus in the study area, but the demand for fruit in all seasons has led to some citrus being cultivated in greenhouses, and agricultural techniques have extended the maturity period of citrus.

Please see the annex for detailed changes (P6,Annotation: Descriptive modifications)

5、In "2.3. Methods", the method description and flowchart were modified. Please see the annex for detailed changes (P7,Annotation: Descriptive modifications)

In addition, we have revised the grammar of the manuscript.

Round 2

Reviewer 2 Report

Although the authors discuss deep learning methods' shortcomings,  DL has been approved as promising in segmentation and detection.  I suggest the authors use DL  in future work. 

Author Response

Thank you for your advice and tolerance. We will continue to work on it. We have continued to make grammatical changes to "Moderate English changes required". We apologize for the trouble we caused you during the reading process. Please see the attachment for details. Thank you for your review!

Reviewer 3 Report

The paper was improved according to my comments

I accept the paper in its current form. However, the list of references should be put in order because the first reference is [15] not [1].

Author Response

Dear Reviewer, We have revised the references as you requested (starting from [15] which may be related to the revision pattern). Moreover, we rechecked the format of the references and the grammar problems. Please see the attached document for details.
